# Deep-Brain Stimulation for Essential Tremor and Other Tremor Syndromes: A Narrative Review of Current Targets and Clinical Outcomes

**DOI:** 10.3390/brainsci10120925

**Published:** 2020-12-01

**Authors:** Christian Iorio-Morin, Anton Fomenko, Suneil K. Kalia

**Affiliations:** 1Christian Iorio-Morin, Division of Neurosurgery, Université de Sherbrooke, 3001, 12e Avenue Nord, Sherbrooke, QC J1H 5N4, Canada; 2Division of Neurosurgery, Department of Surgery, University of Toronto, Toronto, ON M5T 2S8, Canada; fomenkoa@myumanitoba.ca (A.F.); suneil.kalia@uhn.ca (S.K.K.)

**Keywords:** DBS, tremor, essential tremor, Parkinson’s disease, multiple sclerosis, stroke, trauma, Holmes tremor

## Abstract

Tremor is a prevalent symptom associated with multiple conditions, including essential tremor (ET), Parkinson’s disease (PD), multiple sclerosis (MS), stroke and trauma. The surgical management of tremor evolved from stereotactic lesions to deep-brain stimulation (DBS), which allowed safe and reversible interference with specific neural networks. This paper reviews the current literature on DBS for tremor, starting with a detailed discussion of current tremor targets (ventral intermediate nucleus of the thalamus (Vim), prelemniscal radiations (Raprl), caudal zona incerta (Zi), thalamus (Vo) and subthalamic nucleus (STN)) and continuing with a discussion of results obtained when performing DBS in the various aforementioned tremor syndromes. Future directions for DBS research are then briefly discussed.

## 1. Introduction

Tremor is a prevalent symptom associated with multiple conditions, including neurodegenerative diseases (e.g., essential tremor (ET), Parkinson disease (PD), multiple system atrophy and spinocerebellar ataxias), inflammatory diseases (e.g., multiple sclerosis (MS)), drug toxicity (e.g., lithium, sympathomimetics and chemotherapeutic agents), stroke, trauma and many others [1]. While mild tremor can often be controlled medically, many patients have refractory and intractable syndromes that significantly affect their quality of life [2].

The surgical treatment of tremor evolved from a series of serendipitous discoveries that were initially reported in 1953, when Irving Cooper accidently generated an anterior choroidal artery stroke during a pedunculotomy procedure for PD [3]. It was realized that the globus pallidus internus (GPi) lesion led to improvement in bradykinesia and rigidity without creating motor deficits. A subsequent series of chemopallidotomies for PD further highlighted that patients with tremor improvement typically had thalamic involvement (Figure 1) [4]. Thalamotomy was then studied and demonstrated effective for the relief of tremor, achieving control rates between 69% and 91% [5,6,7], at the cost of adverse side effects that could reach an incidence of 67% when bilateral lesions were performed [7,8]. Deep-brain stimulation (DBS) was introduced as a way to reversibly inhibit the ventrolateral thalamus in patients who had previously undergone a contralateral thalamotomy, with the rationale that it would allow a safer treatment of the second side [9]. After the demonstration that DBS could achieve tremor reduction rates similar to a thalamotomy and with a better side-effect profile, even in unilateral cases [10,11], the technique rapidly replaced lesioning procedures as the preferred surgical option for tremor.

Today, DBS is routinely performed for ET and, less frequently, for PD tremor, MS tremor and other etiologies. The goal of this paper is to review the results of DBS for tremor. We will begin by discussing the various targets reported, then analyze results published for different tremor syndromes.

## 2. Tremor Targets

Tremors of various etiology (e.g., ET, PD, MS) have clearly distinct pathophysiological origins [14]. However, the circuits involved appear to converge at the ventrolateral thalamic area. Multiple structures have been targeted, including the ventral intermediate nucleus of the thalamus (Vim), the nucleus ventralis oralis of the thalamus (Vo) and the posterior subthalamic area (PSA).

The Vim nucleus is the classic tremor target for both DBS and thalamotomy procedures (Figure 2). It is a 3 mm-thick layer of cell bodies receiving fibers from the contralateral deep cerebellar nuclei (dentate, interposed and fastigial). Both structures are connected by the dentato-rubro-thalamic tract (DRTT, alternatively named fasciculus cerebellothalamicus or the cerebellothalamic tract), which travels through the superior cerebellar peduncle, mostly decussates in the brachium conjunctivum, passing through and anterior to the red nucleus before ascending into the Vim thalamus and, to a lesser extent, the Vop [15]. The Vim then projects to the ipsilateral motor (M1) and association cortices (the premotor cortex, supplementary motor area and pre-supplementary motor area). The Vim is bordered by the ventralis caudalis (Vc) nucleus posteriorly (receiving sensory information through the medial lemniscus) and the pallidal receiving area (Vop) nucleus anteriorly (receiving pallidal fibers through the thalamic fasciculus (H1 field of Forel)). Immediately ventral to the Vim and Vop lies a white matter region named prelemniscal radiations (Raprl). In a recent tractography study, the Raprl contained (1) the DRTT fibers arriving from the cerebellum en route to the Vim, (2) tractography streamlines connecting the brainstem to the orbitofrontal and prefrontal cortices, (3) streamlines connecting the pallidum to the pedunculopontine nucleus and (4) streamlines directly connected to the motor and premotor cortices through the internal capsule [16]. The Raprl is itself bordered by the medial lemniscus latero-posteriorly on its way to Vc, by the zona incerta (Zi) latero-anteriorly and by the red nucleus medially. The Zi is a shell of cell bodies embryologically derived from the thalamus and extending from the reticulate nucleus of the thalamus [17]. Together, the caudal portion of the Zi, pallidothalamic tracts (thalamic fasciculus (H1) and lenticular fasciculus (H2)) and the Raprl form the posterior subthalamic area (PSA), and the various terms are often used interchangeably in the literature [18]. The different targets are illustrated in Figure 2 and listed in Table 1 with average coordinates and targeting methods.

Successful tremor relief appears to depend on the engagement of the DRTT [26]. This can be accomplished through leads positioned in multiple targets, including the Vim (where the shorter distance to the DRTT predicts lower current needs) [27] and the PSA.

### 2.1. Vim Stimulation

Vim stimulation was introduced in the early 1990s as a replacement for thalamotomies [9]. Because the Vim cannot be visualized using standard imaging protocols, it is usually targeted indirectly using stereotactic atlases (Figure 2). Typical coordinates are provided in Table 1 and are 15 mm lateral to the midcommissural point (or 11 mm lateral from the wall of the third ventricle), 25% of the anterior commissure-posterior commissure (AC–PC) distance anterior to the posterior commissure, and at the level of the midcommissural line [12]. Many authors have attempted to improve targeting methods using tractographic landmarks, such as the Raprl [28], the DRTT [29] or the medial lemniscus and the pyramidal cortico-spinal tract [30]. The generalizability of these approaches remains to be determined through larger multicenter studies. As such, many centers typically confirm indirect targeting with intraoperative electrophysiological assessments [31]. These can take the form of microelectrode recordings (to identify the Vc border, posterior to Vim) and/or sequential macrostimulation (to confirm the optimal site of tremor arrest and side-effect thresholds before DBS electrode insertion). The goal is to position the final DBS electrode in the antero-infero-lateral part the Vim nucleus, at least 2 mm away from the internal capsule (located laterally) and the Vc nucleus (located posteriorly).

Vim stimulation has been shown to reduce tremor in ET, dystonic tremor, PD and MS (see below). In a recent systematic review of 40 studies [32], overall tremor reduction 12 months after unilateral Vim DBS for ET specifically ranged from 53.4 to 62.8% [33,34]. Action tremor was the most responsive symptom, with up to 78.9% reduction. Bilateral stimulation has been shown safe and effective as well, providing increased (and bilateral) tremor relief (overall reduction ranging from 66 to 78%) and better control of axial and voice tremor [32].

Tremor control from Vim stimulation appears to worsen over time [35,36,37,38]. In one series, 73% of patients had experienced waning stimulation benefit within 5 years of implantation [38]. In another longer-term study, tremor reduction changed from 50% at 3 years to 15% at 10 years following DBS implantation [36]. This phenomenon was observed both in the stimulation OFF state (suggesting underlying disease progression) and, to an even greater degree, in the stimulation ON state (suggesting habituation to stimulation). Habituation to stimulation has been documented as early as 10 weeks following initiation of stimulation [39] and could potentially be prevented by weekly (but not daily) changes in the stimulation parameters [40,41].

The most frequent stimulation-induced side-effects include paresthesias, dysarthria and ataxia. Paresthesias occur in up to 45% of patients [42] and can arise from either Vc stimulation posteriorly (producing localized symptoms in the arm, leg or face) or medial lemniscus stimulation inferiorly (producing hemibody symptoms). These effects can usually be eliminated by switching to a more anterior or superior contact, decreasing the current or switching to bipolar configuration [43]. Dysarthria occurs in 11–38.5% of unilateral cases and 22–75% of bilateral cases. Its neurological substrate remains incompletely understood but may include stimulation of the internal capsule fibers laterally, interference with the DRTT inferiorly, a direct effect from Vim stimulation or a combination of these factors [44]. Dysarthria can also be improved by reprogramming, including strategies to limit current spread (decreasing pulse width, decreasing frequency, decreasing amplitude and bipolar stimulation), switching to a dorsal contact or interleaving of two programs to shape the current more dorsally [43]. Often, a trade-off between tremor control and speech fluency must be made, and patients might alternate between two different settings (one optimized for tremor and the other optimized for speech) depending on their activity. Finally, ataxia is a feature of ET that can be unmasked by tremor control. Gait ataxia, in particular, can be deteriorated by supratherapeutic stimulation, although it has not been shown to be worsened by standard therapeutic stimulation compared to the unstimulated state [45].

Analyses of stimulation parameters have consistently shown that ventral contacts are more effective than dorsal ones, and that stimulation below the intercommissural plane (e.g., below the thalamus) appeared even more efficient [23,46]. This led to the investigation of subthalamic targets for tremor relief.

### 2.2. PSA Stimulation

DBS of the PSA encompasses studies of caudal Zi DBS, Raprl DBS, as well as intermediate electrodes affecting both targets non-specifically (Figure 2). In a 2016 meta-analysis of uncontrolled series, the average tremor reduction reported for PSA DBS across all etiologies was 79% [18]. Stimulation of the PSA or Zi (when analyzed as a subgroup) provided statistically significantly superior tremor relief compared to Vim (50% tremor reduction, *p* < 0.001) or STN (60% tremor reduction, *p* = 0.006). A later randomized, double-blind, crossover trial of 13 patients with ET implanted with leads targeting both PSA and Vim through a single trajectory demonstrated a 64% reduction in tremor for the PSA contact versus 50% reduction in tremor for the Vim contact (*p* = 0.086, therefore not statistically significant due to insufficient power) [23]. Long-term outcomes are only available from retrospective series, but one group reported inferior tremor control with PSA DBS at 3 and 4 years despite equivalent short-term results between 6 months and 2 years [47]. In most studies, PSA stimulation appears to require lower parameters than Vim for equivalent tremor relief (4.35 ± 1.42 mA vs. 5.88 ± 2.1 mA, *p* = 0.0006 in the Barbe et al. trial), therefore suggesting that PSA might be a more efficient target [18,23].

Transient insertional effects reported include post-operative dysarthria (22.5% of patients in the largest series) [48] and somnolence (following bilateral Raprl insertion in PD) [49]. Stimulation-induced side effects are typically experienced at lower thresholds than Vim but have the same incidence and quality. They include paresthesia (arising from medial lemniscus stimulation, posteriorly), dysarthria (internal capsule, laterally), diplopia (red nucleus, medially), imbalance/ataxia (DRTT, inferiorly) and depression (limbic STN, anteriorly) [18,23,50].

Together, these findings suggest that PSA DBS (including caudal Zi and Raprl targets) may provide at least equally effective tremor relief than Vim DBS with a similar side-effect profile and possibly more efficient stimulation parameters. A popular stimulation target in this area appears to be 1–2 mm inferior to the AC–PC plane, at the interface between the caudal Zi and Raprl (circle in Figure 2C) [24,51], where fibers in the PSA are the densest, before they progressively fan out as they reach the Vim and then project to cortical areas [16]. While stimulation of both targets likely affect the same network, the PSA’s proximity to the medial lemniscus and internal capsule makes it more vulnerable to off-target side-effects, and this may indeed factor into the observed lower energy parameters after programming is complete. This is less of an issue with DBS, where changes in programming (such as bipolar configurations) can usually resolve the problem [23], but it might explain why the Vim became the preferred target for lesioning (Figure 1) despite historical reports of significant PSA efficacy [52].

### 2.3. Vo Stimulation

The Vo has been targeted by several investigators to reduce tremor in patients classically unresponsive to Vim stimulation, including MS tremor, post-traumatic tremor, post-stroke tremor and Holmes tremor. It is known that receptive fields for proximal joints are more diffusely distributed in the thalamus than distal limbs. Moreover, it has been proposed that these different syndromes might have a pathophysiology that involve pallidal circuits in addition to the cerebello-thalamo-cortical loop involved in ET. As such, anterior extension of the stimulation field to include the pallidal receiving area (Vop) has been suggested. Two approaches have typically been used. The first involves the planned delivery of two leads, with one at the Vim/Vop border (slightly anterior to the standard Vim target), and another at the Vop/Voa border (2–3 mm anterior to the Vim lead) [53,54]. The second strategy involves the delivery of a Voa rescue lead after failed stimulation in the Vim [55]. With the latter approach, in an attempt to limit posterior spread of the current in Vc from the Vim lead as current is increased, Isaacs et al. also replaced the pulse generator with a Medtronic Restore or PrimeAdvanced, which allowed the Voa lead to be assigned an exclusively positive polarity, while the Vim lead was assigned an exclusively negative polarity [55]. This resulted in a true bipolar configuration between both leads, allowing coverage of the thalamus in Voa, Vop and Vim, without current spread in Vc.

The benefit of Vo stimulation remains uncertain. In a randomized, single-blind, pilot study of 12 patients with dual leads for MS tremor, both Vo and Vim stimulation similarly reduced tremor at 3 months. Concomitant stimulation of both targets appeared to improve tremor control slightly further, and no loss of efficacy was observed at 6 months among initial respondents [53]. Another study of seven patients showed that the addition of a Voa lead in ET patients who showed habituation to Vim stimulation provided an additional 16.7% reduction in tremor, relative to Vim stimulation alone [55]. While the literature clearly establishes the feasibility and safety of dual thalamic lead setups, further studies will be required to confirm the utility of Vo stimulation for tremor.

### 2.4. STN Stimulation

The STN is a preferred DBS target for PD. While typically used to relieve bradykinesia and rigidity, it was quickly recognized, along with GPi, to also suppress tremor. Tremor reduction ranges from 40 to 90% (see section on PD tremor below), with a mean of 78% in the latest dedicated series [56]. STN stimulation has also been attempted in ET, where small case series suggested it could be as effective as Vim [57] or caudal Zi [58]. However, cognitive side effects in the first series led the authors to conclude that Vim should still be favored for elderly patients [57], while the lower current required in Zi stimulation in the second series suggested Zi could be a more efficient target [58]. It should be noted that none of these studies accounted for a placebo effect, which was shown to be present in as much as 53% of patients and led to an average of 70% reduction in tremor amplitude in tremor-predominant PD patients [59]. This highlights the need for rigorous trials in this entity. With regard to dystonic tremor, four cases of STN DBS were also reported, and all patients experienced tremor reduction [60,61], although with this small number, further study will be warranted. Therefore, given the paucity of data, STN stimulation should still be viewed as investigational when used for non-PD tremor.

### 2.5. Combined Targets

Because all discussed tremor targets (Vim, Vo, caudal Zi, Raprl and STN) are located in the same area, many authors have suggested lead trajectories engaging multiple targets through different contacts (Figure 3). Possible combinations critically depend on patient anatomy (e.g., cerebral atrophy, ventricle size, location of sulci and cortical veins) as well as the contact span of the electrode used. It is our opinion, when considering such an approach, that the quality of targeting for the main target should never be compromised to hit the second target. When dual targeting is possible, however, it might provide additional programming options, especially for refractory syndromes, such as MS tremor or Holmes tremor.

On a standard Vim approach using an entry point 2.5 cm lateral to the midline and 1 cm anterior to the coronal suture, a single lead can often simultaneously traverse the Vim (on proximal contacts) and the Raprl (on distal contacts). An additional, parallel, anteromedial lead could simultaneously traverse the Voa (on proximal contacts) and Zi (on distal contacts), providing access to four distinct targets using two leads (Figure 3B). In order to engage Vim and Zi or the intermediate PSA target (interface between Zi and Raprl) on the same trajectory (Figure 3C), the entry point must be moved laterally [62,63]. This approach has been used successfully in many cases of ET [62,63] and Holmes tremor [64], although no large trial has confirmed its formal utility. For patients with tremor predominant PD, posterior trajectories targeting both the DRTT and STN [65] or Vim and STN [66] have also been suggested (Figure 3D).

### 2.6. Directional Leads

For all of the discussed targets above, stimulation can usually alleviate tremor without inducing side-effects. When the DBS lead is not directly at the ideal stimulation target, however, more current is required to reach the target. This results in a larger volume of tissue activated which can involve adjacent structures (e.g., Vc nucleus, medial lemniscus and internal capsule) and generate side-effects [67]. Directional leads have been developed to allow eccentric stimulation around the lead’s axis and can theoretically compensate malpositions of up to 1 mm [68]. Current commercial systems typically consist of ring contacts segmented in three sections that can be individually activated, although many other designs have been proposed [69]. Early experience has shown that directional stimulation can increase the therapeutic window while reducing the therapeutic current strength relative to omnidirectional stimulation [70,71,72,73,74]. Whether or not this translates into improved patient outcome remains unclear and might depend on the underlying condition and target. It was estimated that 40% of patients with thalamic DBS (including ZI and PSA) and 48% of patients with GPi DBS might experience side-effects that limit the therapeutic benefit, compared to 12% of STN DBS patients [75], and the former might therefore benefit most from directional stimulation.

Programming complexity exponentially increases with each additional contact available. As such, many centers that implant directional leads program the segmented contacts as a single, omnidirectional system. Directionality is used only as a backup when side-effects warrant further exploration of programing options. Rational programming is also more challenging, as the orientation of directional leads is difficult to interpret on imaging and requires rotation fluoroscopy or specialized software to interpret CT artifacts, such as the Directional Orientation Detection (DiODe) algorithm [76]. It has been shown that the final position of the leads often differs from the intended orientation at implantation, [77,78] and thus needs to be validated postoperatively. Analysis of local field potentials from directional systems can be used to predict the best stimulation contact [79,80,81], although until systems combining both sensing capabilities and directional leads are available, this approach remains experimental.

## 3. Tremor Syndromes

The various targets discussed above have been applied to multiple tremor syndromes, which we will now discuss.

### 3.1. Essential Tremor

Essential tremor is seen as neurodegenerative disorder of the cerebellum in which an unknown trigger leads to loss of Purkinje cells [82]. This reduces the GABAergic input to cerebellar output nuclei, inducing excessive cerebellar output to the thalamus. Combined with a chronic reorganization or the remaining Purkinje cell synapses in the cerebellar cortex, oscillations arise in motor networks that generate an action tremor [83]. Most patients with ET initially respond to propranolol or GABAergic agents, such as topiramate, gabapentin, primidone and ethanol [84]. As the disease progresses, patients may become refractory to medication, resulting in a reduced quality of life, and may be considered for surgical intervention. DBS in ET is thought to act by inhibiting the pathological oscillations in the cerebello-thalamic pathways. The various targets discussed above are all nodes in this network that have been targeted and shown to reduce action tremor in ET (see above).

When considering all targets together, a meta-analysis was recently performed and identified 37 studies (14 retrospective and 23 prospective) published between 1996 and 2019 and reporting the results of 1202 patients treated with DBS for medically refractory ET [85]. The average improvement in tremor score was 60.1% ± 9.7 (SD) at an average follow-up of 16.6 months. Tremor reduction was superior in patients receiving bilateral DBS (61.2% ± 5.2, SD, *n* = 69) versus unilateral DBS (56.4% ± 9.7, SD, *n* = 472, *p* < 0.001). Across studies, DBS for ET improved quality of life by 52.5% ± 16.2 (SD) at an average follow-up of 16.6 months [85]. Perioperative and hardware-related complications included lead problems (11.4%), local adverse symptoms (10.9%), infection (1.8%), intracranial hemorrhage (1.1%), seizures (0.5%) and death (0.1%). Stimulation-related side-effects included speech disturbance (11.1%), gait disturbance (10.9%), paresthesias (8.8%), neuropsychiatric issues (5.1%) and other symptoms (8.3%). Across all complications, 76.6% were transient [85]. Overall, patient satisfaction with DBS for ET is very high [86], and outcomes appear to have been slightly improving over time (average of 0.4% improvement in tremor score per year in studies published between 1996 and 2019, Spearman’s correlation coefficient = 0.357, *p* = 0.035), suggesting progress in the surgical techniques and hardware [85].

Many challenges remain in this field. Habituation to stimulation, which occurs in 73% of patients within 5 years of implantation [38], is a significant concern to which there is yet to be a convincing solution. The complexity and cost of DBS for ET may also be challenged as thalamotomies are being revisited through the introduction of less invasive lesioning modalities such as radiosurgery and focused ultrasound, both of which may trend to being more cost-effective than DBS [87,88]. However, these types of analysis may be difficult to interpret, as robust long-term data for lesions and recurrence of tremor are not as clear in large cohorts. This also needs to be weighed against new devices with longer battery life and increased programming options. The demonstrated safety of bilateral thalamic DBS currently maintains neuromodulation’s edge over lesions, although this may change when current ongoing studies of modern bilateral lesions are completed (NCT04501484 and NCT03465761). If these studies are successful, the main question for the field will be the long-term benefit and frequency of complications with each approach.

### 3.2. Dystonic Tremor

Dystonic tremor (DT) is a tremor syndrome “combining tremor and dystonia as the leading neurological signs” [1]. There is considerable overlap with ET and ET plus syndromes, and many patients are often misclassified in these categories, which may represent a continuum [89]. The pathophysiology of dystonic tremor is even less understood than ET, but might involve similar cerebellar pathology, in addition to dystonia-specific loss of inhibition in spinal and brainstem circuits [90].

In the largest series of Vim DBS for DT, 26 patients followed for more than 6 months showed a 44–56% improvement in tremor score sustained over 5 years [91]. Tremor control was not inferior to ET patients similarly treated at the same institution. Functional status was improved by 56% after 6 months and 1 year, but did not reach statistical significance past 2 years, suggesting suboptimal control of the concomitant dystonia symptoms might be limiting activities of daily living over the long term. This might be relieved by GPi DBS, which has known potency against dystonic symptoms, although the only series comparing tremor control from GPi and Vim DBS in 10 DT patients found GPi inferior to Vim [92]. As for STN DBS for dystonic tremor, only four cases were reported, but all were successful [60,61]. Larger trials will be needed to clarify the optimal target. When considering performing DBS for DT, the involvement of a multi-disciplinary team may be important to debate target selection.

### 3.3. Parkinson’s Tremor

Resting and action tremors affect 76–100% and 47–90% of PD patients, respectively, and are often the most visible symptom of the disease [93]. While DBS in PD is usually performed to alleviate bradykinesia and rigidity, concomitant tremor reduction has consistently been observed. The pathophysiology of PD tremor is poorly understood but might involve loss of segregation of parallel basal ganglia circuits, independent thalamic pacemakers or pathological coupling of cerebellothalamocortical circuits and the basal ganglia [14].

A meta-analysis of 5 randomized controlled trials including 489 patients demonstrated a 23–100% reduction in tremor at various time points, with no overall difference in efficacy between STN and GPi targets [93]. In the off-medication state at the follow-up closest to 12 months, tremor reduction was 37–89% for STN and 25–79% for GPi. While many authors have reported that tremor suppression from GPi stimulation might be delayed by days to week, this target was found to have a more stable postoperative efficacy compared to STN, which is more effective at 12 months than at 6 months after implantation. STN was also found to be equally effective as Vim [94]. Improvement can be seen in both tremor-predominant and akinetic-rigid variants. Most importantly, the tremor’s responsiveness to medication is not predictive of successful tremor control with stimulation and can be achieved (>25% reduction) in 94% of cases [56]. Arm and chin tremor might be more responsive to DBS than leg tremor.

### 3.4. MS-Associated Tremor

MS-associated tremor appears to be less responsive to DBS than the previously discussed syndromes, although it can still be improved. In a meta-analysis of 13 studies including 129 DBS patients, stimulation improved the Hedges standardized mean tremor score by 2.15 [95]. This effect size is difficult to transpose into clinical scales, as it is a pooled estimate from heterogeneous outcome measures, but the largest study of combined Vim and Vo DBS demonstrated a 29.6% tremor reduction [53], while series of PSA DBS showed a 50–60% improvement on average [18]. The number of cases is too small to compare targets to one another, and the low rate of DBS implantation for MS tremor, even in high-volume centers, makes it unlikely that large trials will be conducted in the near future. For now, some authors have suggested using a dual lead/dual target strategy (Figure 3B) to maximize the chance of good outcome [53], although the efficacy of this approach remains empirical. Careful discussion of the goals of the treatment with the patient is critical to manage expectations in these complex tremor cases.

### 3.5. Other Tremors

DBS has been used in a variety of rarer indications, including post-stroke tremors, lesion-related tremors and post-traumatic tremors. Many of these cases also fulfill the criteria for Holmes tremor, which consists of a slow (usually <5 Hz) rest, postural and intention tremor typically appearing up to 2 years following central nervous system injury [1,96]. The literature on these syndromes is limited to small case reports and case series providing anecdotal evidence of efficacy. In a recent meta-analysis of 35 studies reporting 82 patients, Vim DBS was performed in 63.6% of cases, GPi in 18.2% and other targets (PSA and Vo) in the remaining cases [96]. Median tremor improvement was 75%, although a significant publication bias might be present. Post-stroke tremor appeared to have better prognosis than post-traumatic syndromes. Reports have also been published for orthostatic tremor [97], tremor with underlying ataxia syndromes [98], Wislon’s disease-associated tremor [99], Klinefelter’s syndrome [100] and fragile X-associated tremor ataxia syndrome [101]. The reader is referred to these individual publications for further consideration of these rare indications.

Tremor reduction scores derived from the largest series or meta-analyses available. When multiple timepoints are reported, the data closest to 12 months were used. Shaded cells represent combinations for which reliable data or meta-analyses are unavailable. Given the large methodological differences between studies and different patient populations, these numbers cannot demonstrate the superiority or inferiority of one target over another.

## 4. Future Directions and Conclusions

DBS for medically refractory tremor has reached a state of maturity. While its efficacy in ET and PD is well established (Table 2), the mechanisms underlying tremor suppression, stimulation habituation and side effect generation are still incompletely understood and will continue to be researched. Incremental improvement in surgical techniques has already been shown to have slowly improved outcomes over the last two decades [85]. Further innovations in hardware are likely to extend battery life, minimize hardware-related complications and optimize stimulation conformity to the desired target. Target selection will continue to be further refined, hopefully soon reaching patient-specific anatomical targeting based on advanced imaging of the pathological network. This should provide a rational way to further improve outcomes, especially for rare indications (MS, stroke and trauma) where large-scale randomized controlled trials are unrealistic. However, until disease-modifying strategies emerge, DBS will remain an invasive, temporizing measure offering improved quality of life, but still ineffective against the underlying progressive neurodegenerative processes. With directionality and future closed-loop stimulation paradigms on the near horizon, there is a positive outlook on the potential to address challenges with habituation and thresholds for off-target side effects with disease progression. The rebirth of thalamotomies using less invasive approaches such as radiosurgery and focused ultrasound may further challenge DBS’ role in an increasingly cost-constrained environment. Lastly, the recent suggestion that DBS might have neuroprotective effects [102] opens a new world of possibilities currently unachievable by lesioning modalities. Leveraging this unique capability will require new study designs that focus less on the targets themselves and more on how to mitigate the loss of neurons within the circuit. Helping a patient with tremor navigate the plethora of options will generate new complexities for physicians but, importantly, can empower patients to decide which surgical intervention is best for them.

## Figures and Tables

**Figure 1 brainsci-10-00925-f001:**
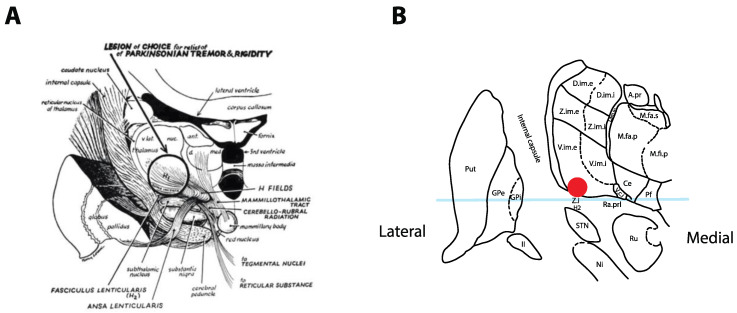
Historical tremor target. (**A**) Coronal representation by Cooper in 1960 of the ideal lesioning target for the control of tremor. Note how the lesion encompasses the ventrolateral thalamus in addition to the zona incerta (Zi) and posterior subthalamic area (PSA). Reproduced from [4] with permission. (**B**) Optimal location for thalamic stimulation based on current data [12] plotted on a coronal representation of the thalamus, 5 mm posterior to the midcommissural point [13]. The light-blue overlay represents the midcommissural plane.

**Figure 2 brainsci-10-00925-f002:**
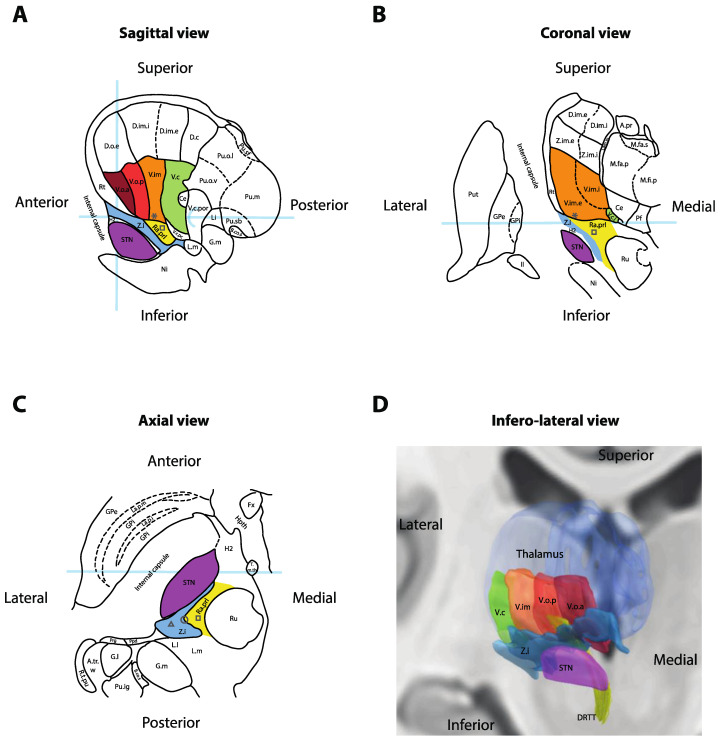
Anatomy of the ventral thalamic area. (**A**) Sagittal cross-section of the thalamus, 14.5 mm lateral to the midline. Light-blue overlays represent the intercommissural plane (horizontally), the midcommissural plane (vertically) and the posterior commissure (half-circle). (**B**) Coronal cross-section 5 mm posterior to the mid-commissural point. Light-blue overlay represents the intercommissural plane. (**C**) Axial cross-section 3.5 mm inferior to the intercommissural plane. Light-blue overlay represents the midcommissural plane. Panels A, B and C redrawn and adapted from [13]. (**D**) Three-dimensional reconstruction of the right thalamus seen from an antero-infero-lateral oblique view. Nuclei rendered in Lead-DBS v2 [19] using the DISTAL atlas (ventralis caudalis (Vc), ventral intermediate nucleus of the thalamus (Vim), pallidal receiving area (Vop), Voa and subthalamic nucleus (STN)) [20], Zona incerta atlas (Zi) [21] and Brainstem connectome atlas (DRTT) [22]. * = Standard Vim target for deep-brain stimulation (DBS); square = standard prelemniscal radiations (Raprl) target for DBS; triangle = standard caudal Zi target for DBS; circle = intermediate PSA target at the interface between Zi and Raprl; II = optic tract; A.pr = nucleus anteroprincipalis thalami; A.tr.w = area triangularis (Wernicke); B.co.s = brachium colliculi superioris; Ce = nucleus centralis; D.c = nucleus dorso-caudalis; D.im.e = nucleus dorso-intermedius externus; D.im.i = nucleus dorso-intermedius internus; D.o.e = nucleus dorso-oralis externus; DRTT = dentato-rubro-thalamic tract; Fx = fornix; G.l. = corpus geniculatum laterale; G.m = corpus geniculatum mediale; GPe = globus pallidus externus; GPi = globus pallidus internus; H2 = fasciculus lenticularis; Hpth = hypothalamus; La.m.ip = lamina medialis interpolaris; La.p.i = lamina pallidi incompleta; La.p.m = lamina pallidi medialis; Li = nucleus limitans thalami; L.l = lemniscus lateralis; L.m = lemniscus medialis; M.fa.p = nucleus medialis fasciculosus posterior; M.fa.s = nucleus medialis fasciculosus superior; M.fi.p = nucleus medialis fibrosus posterior; Ni = substantia nigra; Pf = nucleus parafascicularis thalami; Ppd = nucleus peripeduncularis; Prg = praegeniculatum; Pu.ig = nucleus pulvinaris intergeniculatus; Pu.m = pulvinar mediale; Pu.o.l = nucleus pulvinaris orolateralis; Pu.o.v = nucleus pulvinaris oroventralis; Pu.sb = nucleus pulvinaris suprabrachialis; Pu.sf = nucleus pulvinaris superficialis; Put = putamen; Ra.prl = radiatio praelemniscalis; Rt = reticulatum thalami; R.t.pu; Ru = red nucleus; STN = subthalamic nucleus; T.m.th = tractus mammillo-thalamicus; V.c = nucleus ventrocaudalis; V.c.i = nucleus ventrocaudalis internus; V.c.pc = nucleus ventrocaudalis parvocellularis; V.c.por = nucleus ventrocaudalis portae; V.im = nucleus ventrointermedius; V.im.e = nucleus ventrointermedius externus; V.im.i = nucleus ventrointermedius internus; V.o.a = nucleus ventrooralis anterior; V.o.p = nucleus ventrooralis posterior; Z.i = zona incerta; Z.im.e = nucleus zentrolateralis intermedius externus; Z.im.i = nucleus zentrolateralis intermedius internus. * = Standard Vim target for DBS; square = standard Raprl target for DBS.

**Figure 3 brainsci-10-00925-f003:**
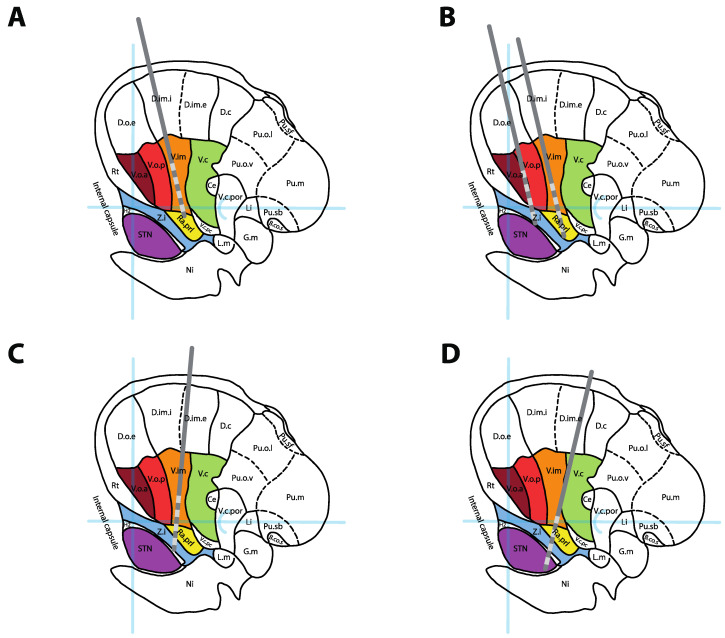
DBS lead positions for single and combined targeting. (**A**) Standard position of a Vim DBS lead. (**B**) Dual thalamic leads setup targeting Vo + Zi (anterior lead) and Vim + Raprl (posterior lead). (**C**) Combined targeting of Vim + PSA/Zi in a single trajectory. (**D**) Combined targeting of the DRTT (in Raprl) + STN in a single posterior trajectory. All leads represented over a sagittal cross-section 14.5 mm lateral to the midline, adapted from [13]. See Figure 2 legend for abbreviations.

**Table 1 brainsci-10-00925-t001:** Coordinates and anatomical targeting of common tremor targets.

Target	Indirect Coordinates	Anatomical Targeting
Vim	X = 15 mm lateral to MCP (or 11 mm lateral from wall of third ventricle) Y = 25% AC–PC distance posterior to MCP Z = level of MCP	N/A
PSA (intermediate target) [23]	X = 11 ± 1.36 mm lateral to MCP Y = 5.65 ± 1.42 posterior to MCP Z = 1.87 ± 0.62 mm inferior to MCP	Halfway between the lateral border of the red nucleus and the medial border of the STN, on a line perpendicularly intersecting the STN axis [24].
Raprl [25]	X = 11.63 ± 0.66 mm lateral to MCP Y = 6.73 ± 1.62 mm posterior to MCP Z = 4.38 ± 1.02 mm inferior to MCP	Perirubral white matter
Caudal Zi [25]	X = 14 ± 1.56 mm lateral to MCP Y = 5.8 ± 1.46 mm posterior to MCP Z = 2.1 ± 1.05 inferior to MCP	Behind posterior end of STN

MCP = Midcommissural point; AC-PC = Anterior commissure-Posterior commissure distance.

**Table 2 brainsci-10-00925-t002:** Tremor reduction after DBS surgery stratified by disease and target.

	Vim	PSA (ZI/Raprl)	Vo	STN	Other Target	All Targets
Essential tremor	53–63% [32]	62–80% [32]				60% [85]
Dystonic tremor	44–56% [91]				GPi: 50% [92]	
Parkinson’s tremor	72% [94]			37–89% [93]	GPi: 25–79% [93]	
Multiple sclerosis-associated tremor	21% [53]	50–60% [18]	20% [53]		Combined Vim + Vo: 30% [53]	
All indications	50% [18,23]	64–79% [18,23]		60% [18]		

Greyed out cells represent conditions for which reliable data is unavailable.

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
