# Peer review of "Deep-Brain Stimulation for Essential Tremor and Other Tremor Syndromes: A Narrative Review of Current Targets and Clinical Outcomes"

_brainsci, 2020, doi:10.3390/brainsci10120925_

Round 1

Reviewer 1 Report

excellent and thorough review of the topic.

Minor points- Possible increased discussion of multi-segmental leads could make review more current.

Would add STN stim for dystonic tremor( ther is a small literature on this topic)

Author Response

We thank the reviewer for his insightful comments.

We added a complete section on directional leads.

A discussion of STN DBS was added to the dystonic tremor section.

Reviewer 2 Report

This review article seeks to describe current approaches to treating tremor associated with multiple diseases with DBS, and their resulting outcomes. The article is well written and informative, especially from the neurosurgical/neuroanatomy standpoint. I only have a few relatively minor suggestions for the authors, as follows  
  • In some parts, the manuscript relies heavily on previous review articles instead of novel research reports (see, e.g., the discussion of VIM DBS outcomes on page 5, relying on the review paper by Dallapiazza et al). I would suggest including citations of the original research articles as well.
  • The manuscript is somewhat repetitive in parts (e.g., discussion of stimulation habituation in the VIM DBS section and the ET section). I would suggest possibly re-organizing the document such that diseases are introduced first with non-DBS therapies, and then responses to stimulation are included in the discussion of various targets, possibly with subsections for outcomes in each disease.
  • The sections on individual tremor syndromes have varying levels of detail on pathophysiology vs outcomes (only the ET section appears to have significant information on the underlying pathology) - I would suggest adding some background to each section on the underlying disease state.
  • While directional DBS is presented as "on the horizon", there exist more than a few studies over the last few years looking at this technology in both PD and ET. This information should probably be included, as it is an important evolution in the treatment of these diseases and may have disparate effects on tremor.

Author Response

We thank the reviewer for his insightful comments.

  • More primary literature is now cited in addition to the Dallapiazza review. Because citing every study included in this meta-analysis would have involved including over 50 new references, we decided to cite most important studies (i.e. either the ones with the largest n, or the studies defining the ranges (min and max) of each outcomes). These are now included in addition to the meta-analysis which provided the average we report.
  • We shortened the repetitive sections to keep targets as disease-agnostic as possible and vice-versa.
  • We added background on the pathophysiology of each tremor syndrome for each disease section.
  • We added a new section on directional leads discussing the advantages and the challenges associated with their use.

Reviewer 3 Report

In this useful review, authors summarize the current state of art of deep-brain stimulation for tremor. Their contribution help to understand which brain area are involved in tremor phenomenon and consequent effects of DBS application in multiple tremor syndromes.

The paper is well written and provides a good overview of the different approaches that are being used in this field. The only comments that I would make are:

- P 2.4:  it would be useful to provide a bit more detail on whether studies use  deep brain stimulation to investigate the effect of placebo regarding subthalamic nucleus –DBS. Barbagallo et al.(https://doi.org/10.1016/j.parkreldis.2018.03.012), discussed about the assess the placebo-effect on resting tremor citing some studies about DBS applied to STN.

- P 3: similarly, it would be useful to provide summary tables of the key studies mentioned in Section 3 and their findings for the non-expert reader to refer to. 

Author Response

We thank the reviewer for his insightful comments.

  • The placebo effect was discussed and the suggested study referenced in the STN DBS section.
  • A new Table 2 was created to summarize the reported tremor control rates for each disease and target. Because of wide methodological differences between studies, this Table should be used with caution and cannot be used to directly assess the superiority of one target over another. This limitation is stated in the table legend.